# Improved U-Net with Residual Attention Block for Mixed-Defect Wafer Maps

**Jaegyeong Cha** and **Jongpil Jeong** *

Department of Smart Factory Convergence, Sungkyunkwan University, Suwon 16419, Gyeonggi-do, Korea;
sean9887@naver.com
* Correspondence: jpjeong@skku.edu

**Abstract:** Detecting defect patterns in semiconductors is very important for discovering the fundamental causes of production defects. In particular, because mixed defects have become more likely with the development of technology, finding them has become more complex than can be performed by conventional wafer defect detection. In this paper, we propose an improved U-Net model using a residual attention block that combines an attention mechanism with a residual block to segment a mixed defect. By using the proposed method, we can extract an improved feature map by suppressing irrelevant features and paying attention to the defect to be found. Experimental results show that the proposed model outperforms those in the existing studies.

**Keywords:** segmentation; mixed-type wafer maps; U-Net; attention mechanism

## 1. Introduction

Semiconductor chips, which are used in industries and electronic products that we often talk about, are semiconductor integrated circuits. Semiconductor wafers can be manufactured by eight different processes [1], but these are largely divided into front-end and back-end processes. The front-end process is used to design a semiconductor chip and engrave it on the wafer; the back-end process is used to cut the chip engraved on the wafer and wrap it with an insulator to lay wires to receive power stably [2,3]. In particular, in the front-end process, also called the wafer process, a single semiconductor chip is made by repeatedly forming and cutting various types of films on the wafer surface to create an electronic circuit [4]. Techniques such as Photolithography, which transfers semiconductor circuit patterns onto a wafer; Etching, which cuts off parts other than circuit patterns; Deposition, which forms an insulating thin film for separation and protection between metal, circuitry, and metal for transmitting electrical signals; and Metallization, which forms wiring, are included in the previous process [5,6]. Because of these various processes, the types of defects that occur on the wafer are also diverse. After wafer fabrication, several tests are performed to identify defects and represent them as binary values on the wafer map. In this way, the classification results for the dies on the wafer map form a specific pattern and are visually displayed [7]. The various defect patterns on the wafer map are related to the manufacturing process. Therefore, accurately classifying the defect patterns on the wafer map allows defect sources from the manufacturing process to be identified. Such classification is also important because it provides engineers with clues for solving problems [8].

Because of recent advances in miniaturization technology and an increase in wafer size, the probability of generating two or more mixed defect patterns has increased [9,10]. Detecting defect patterns is more complex, because mixed defects can have many combinations of causes, such as location, size, type, and number. Recently, because of the development of artificial intelligence, research based on deep learning is being widely conducted. Kyeong and Kim [11] proposed using convolutional neural networks to classify

mixed defect patterns, and Wang et al. [12] proposed deformable convolutional networks. Kim et al. [13] proposed an infinite warped mixture model for clustering mixed defect patterns. Ming-Chuan Chiu et al. [14] proposed data augmentation and mask R-CNN for instance segmentation in mixed-type defect patterns.

As a method for detecting defects, segmentation is used one step further from classification. Classification simply classifies the target image, while segmentation can infer data on a pixel-by-pixel basis. This can greatly aid in decision making by providing additional information to the user. In particular, many segmentation studies are being conducted for the purpose of finding diseases in the medical field. Ozan Oktay et al. [15] proposed a model applying an attention gate to U-Net for medical image segmentation. Through the attention gate, the model learns by automatically focusing on target structures of various shapes and sizes. Xiaocong Chen et al. [16] proposed a novel U-Net architecture using aggregated residual blocks and a soft attention mechanism for segmentation of regions infected with COVID-19. Yu-Cheng Liu et al. [17] proposed a cascaded atrous dual-attention U-Net for accurate tumor segmentation. They introduced a cascade structure to extend low-resolution quality prediction, and proposed a dual-attention module to improve the functional expression of tumor segmentation.

In this study, the residual attention block and the integrated residual block and attention module were combined with U-Net to provide engineers with segmentation results focused on the defect area. We make specific contributions as follows:

1.　Applying an attention-guided U-Net for classification of wafer defects.
2.　Reducing unnecessary human resources and time by generating the ground truth essential for training the segmentation model with an automatic defect masking technique.
3.　Performing detection of mixed faults using only a single fault using the training of the proposed model.

The structure of the paper is as follows. Section 2 describes related work. Section 3 describes in detail the architecture and features of the proposed model. Section 4 describes the experimental procedure and results. Finally, Section 5 presets the conclusions and suggestions for future research.

## 2. Related Work

### 2.1. Semiconductor Wafer Map

A semiconductor wafer is a circular plate in which a single crystal pillar made by growing silicon (Si), gallium arsenide (GaAs), etc., as a core material for semiconductor integrated circuits is sliced to an appropriate thickness. A semiconductor integrated circuit is an electronic component in which many devices are integrated into a single chip to process and store various functions. In other words, the wafer is the basis of the semiconductor, because the semiconductor integrated circuit is elastic by making the circuit on a thin circular plate, that is, a wafer.

Wafer mapping is widely used for data analysis in semiconductor manufacturing processes. Wafer mapping creates a map in which the performance of the semiconductor device on the wafer surface is color-coded according to the test result of each chip defect. The generated wafer map has one or several patterns that depend on the distribution of defective chips. Since the defective chip pattern is formed differently depending on the cause of the abnormality in the semiconductor process, various defect patterns are generated. Therefore, the analysis of defective chip patterns, that is, wafer map defects, provides important information for detecting abnormalities in the semiconductor process and identifying the causes of defects. Figure 1 shows the wafer map.

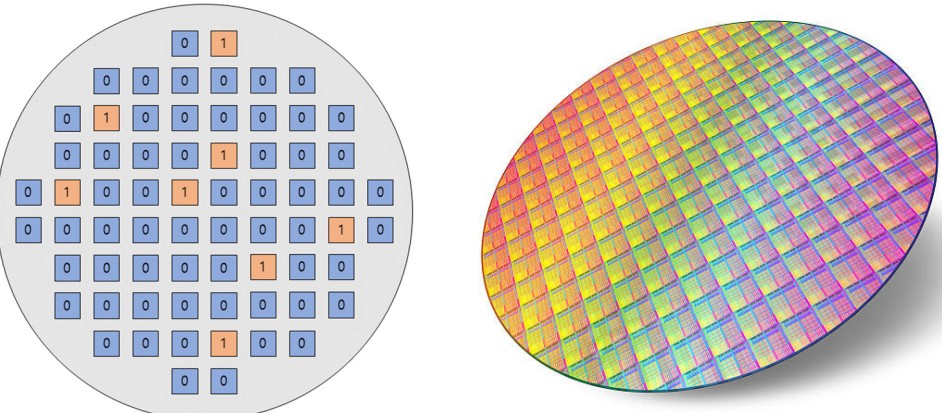

**Figure 1.** Example of wafer map.

### 2.2. U-Net

Semantic segmentation is the task of classifying objects in pixels in an image [18]. In deep learning, an object means a class to be classified [19]. Segmentation needs to obtain a label from the input image that each pixel belongs to. For this, one-hot encoding is used to create as many output channels as there are classes. After that, one output is calculated by means of argmax. In this way, semantic segmentation considers only whether pixels are binarily included for each class. It is mainly studied in the medical and transportation fields and has been recently applied to autonomous vehicle technology. Representative studies include the fully convolutional networks proposed by Jonathan Long et al. [20] in 2015, DeepLab proposed by Liang-Chieh Chen et at. [21] in 2015, and U-Net proposed by Olaf Ronneberger et al. [22].

U-Net is a model based on an end-to-end fully convolutional network (FCN) proposed for the purpose of biomedical image segmentation. U-Net uses data augmentation to learn enough with only a small amount of data. Data augmentation improves model performance by increasing the amount of data and is important because it simply improves most of the model performance [23,24]. U-Net is shaped like a U and mainly consists of a contracting path and an expansive path. The contracting path is the stage that captures the context of the input image. Context refers to the relationship between neighboring image pixels and can be thought of as understanding the overall image context by looking at a part of the image. In general, it consists of a convolutional layer and a pooling layer. In each step, the number of channels in the feature map is doubled, but the size is cut in half. The expanding path is a step for accurate localization, and up-sampling is performed several times to increase the resolution that has gone through the contracting path. It plays a role in more accurate localization by combining with the context of the feature map captured in the contracting path. U-Net removes the fully connected layer and uses only convolution to enable segmentation even when images of arbitrary size come in by the overlap tile strategy. Overall, the contracting path and expanding path are symmetrical and constitute a U-shaped network.

### 2.3. Attention Mechanism

Attention mechanism is a technique to deal with the fact that the translation quality deteriorates as the input sentence lengthens in the field of machine translation [25]. The basic idea is that every time the decoder predicts the output word, it once again consults the entire input sentence at the encoder. The concept of attention in the convolutional neural network (CNN) series has been mainly used in feature selection using multi-modal relationships such as image captioning [26,27]. Attention itself is 'attention' on a certain characteristic, and in image classification or detection problems, it is necessary to focus on an important part according to the input image.

In 2018, Jongchan Park et al. [28] proposed two self-attention modules to improve the performance of CNNs. The Bottleneck Attention Module (BAM), as the name suggests, is located at the bottleneck of each network, which is the part where spatial pooling takes place. Spatial pooling is an essential part of the CNN abstraction process, and the spatial resolution of the feature map becomes smaller. A key feature of BAM is to increase the value of the important part with attention and to decrease the value of the less-important part by adding the BAM before the amount of information decreases in this section. BAM takes 3D conv features as input and output conv features refined with attention. The attention of the channel axis and the spatial axis is divided and calculated, each output value is added, and a 3D attention map of the same size as the input is generated by means of the sigmoid. The channel axis collects the global context of each channel through global average pooling. It then passes through the MLP and outputs the same size as the input channel. The spatial axis calculates the final 2D attention only by convolution in order to maintain the meaning of the channel. BAM shows significant performance improvement without significantly increasing the parameters and computations of the existing backbone.

Convolutional Block Attention Module (CBAM) is a follow-up study of BAM [29]. BAM is implemented by adding channels and spatial to one 3D attention map, but CBAM works better by sequentially applying channels and then applying spatial. Channel attention utilizes the internal channel relationship of input feature F to generate a channel attention map. Channel attention focuses on what is important in a given input. To compute effectively, it compresses the spatial dimension of the input feature map to $1 \times 1$, and it also applies average pooling and max pooling to incorporate spatial information. Using the two pooling operations together improves performance. Since the two pooled features are values that share the same meaning, one shared MLP can be used, and the number of parameters can be reduced. For spatial attention, we compute spatial attention with only one conv. The difference between spatial attention and channel attention is that spatial attention focuses on 'where' information is. Spatial attention concatenates two values generated by applying max pooling and average pooling to the channel axis in the feature map generated by multiplying the channel attention map and the input feature map. A $7 \times 7$ conv operation is applied here to generate a spatial attention map. As such, the BAM and CBAM network structures consist of simple pooling and convolution, and self-attention is modularized so that it can be easily incorporated into any deep learning model.

## 3. Improved U-Net with Residual Attention Block

### 3.1. Network Architecture

In this paper, we propose a U-Net using a residual attention block for segmentation of semiconductor wafer maps that include mixed pattern defects. Inspired by the residual block and attention module (CBAM), we integrated the residual attention block into the U-Net architecture. The proposed architecture is shown in Figure 2.

### 3.2. Residual Attention Block

Figure 3 shows the structure of the residual attention block. Residual block is a structure used in ResNet as designed by Kaiming et al. [30]. It is a simple idea that takes the input as it is and adds it to the learned function as a method designed to solve the problem of inferior performance caused by the gradient vanishing as the layer of the model gets deeper.

In this study, we used a residual block for each step of the contracting path and the expanding path to solve the problem of degradation of the model's performance, which can help the model to extract more features from every layer. This process can be expressed as:

$$x_{i+1} = x_i + F(x_i) \tag{1}$$

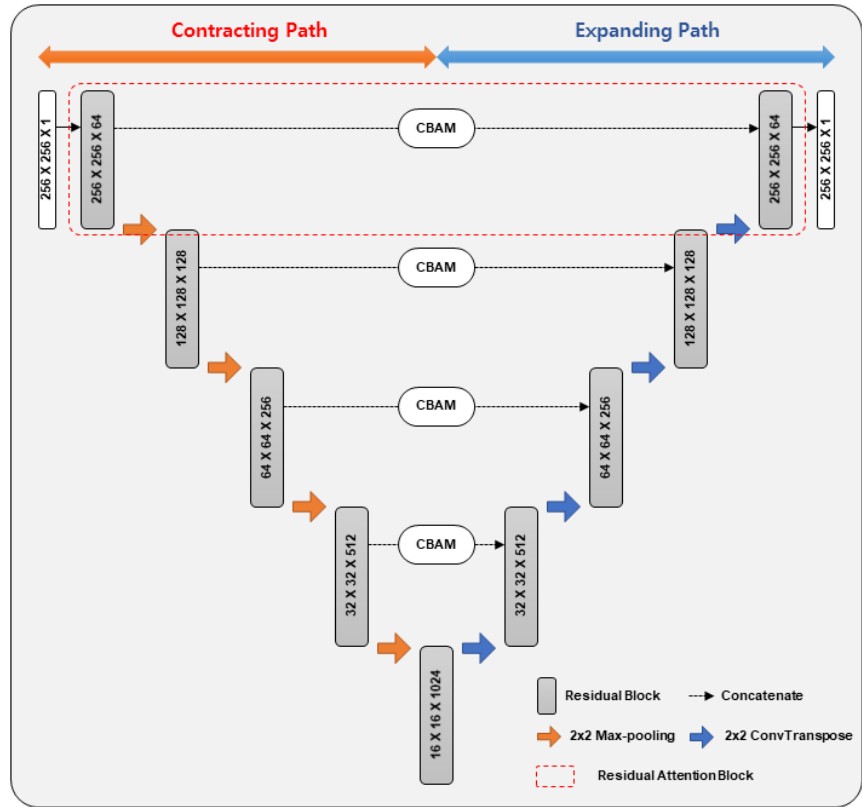

**Figure 2.** Overall architecture of proposed model.

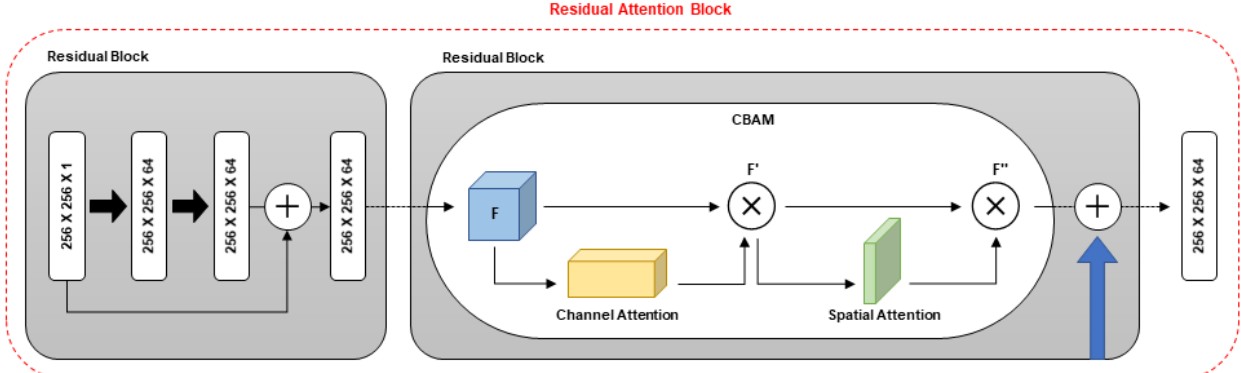

**Figure 3.** Residual attention block.

U-Net concatenates the feature map generated from the contracting path and the feature map generated from the expanding path step by step through skip connection. By combining context information and location information of the shallow and deep layers, important information in the image can be conveyed well, and a clearer image can be obtained to make accurate predictions. We tried to improve the performance of the model by using the attention module to focus on a specific area of the image.

CBAM sequentially applies channel attention and spatial attention as shown in Figure 2. This process can be expressed as:

$$F' = A_c(F) \bigotimes F \tag{2}$$

$$F'' = A_S(F') \bigotimes F'' \tag{3}$$

Channel attention uses a combination of max pooling and avg pooling; when the input feature *F* is input, channel attention $A_c$ is output.

$$A_c(F) = \sigma(MLP(AP(F)) + MLP(MP(F))) \tag{4}$$

In Equation (4), AP means average pooling, MP means max pooling and $\sigma$ means batch normalization. Spatial attention outputs spatial attention $A_S$ when the refined $A_c$ in the previous channel attention is input. If channel attention learns intensively on 'what,' spatial attention learns intensively on 'where.'

$$A_s(F) = \sigma\left(f^{7\times7}([AP(F); MP(F)])\right) \tag{5}$$

The output result of the attention module is concatenated with the up-sampling feature map of the output of the expanding path one step before. This process enables segmentation to be improved by focusing on the area of the defect to find contextual information and location information of the shallow and deep layers.

### 3.3. Contracting Path

The contracting path consists of four residual blocks and four $2 \times 2$ max pooling layers. The input image is a $256 \times 256$ resolution black-and-white image with one channel. The residual block consists of two $3 \times 3$ convolution layers, batch normalization following each layer, and ReLU activation functions. In the first residual block, the $256 \times 256 \times 1$ input image is extracted as a $256 \times 256 \times 64$ feature map. Then, the size of the feature map is halved, and the number of channels is doubled by the down-sampling of the max pooling layer, which then has a $32 \times 32 \times 512$ feature map of four residual blocks through the contracting path.

The bottleneck is the transition section between the contracting path and the expanding path. It consists of one residual block and has a $16 \times 16 \times 1024$ feature map. The output of the bottleneck goes into the contracting path.

### 3.4. Expanding Path

The expanding path has four residual blocks, as does the contracting path. However, since the size of the feature map reduced in the contracting path needs to be restored, a $2 \times 2$ transposed convolution layer is used instead of the max pooling layer to perform up-sampling. Therefore, in each expanding layer, the size of the feature map is doubled, and the number is halved. The feature map up-sampled in the previous layer is concatenated with the output of the attention module. As a result, it has a feature map of $256 \times 256 \times 64$ size by means of four residual blocks through the expanding path. Finally, by means of a $1 \times 1$ convolution layer, each pixel generates a vector representing information about the corresponding class.

### 3.5. Loss Function

In this study, we used the cross-entropy loss function as the loss function of the proposed network [31]. The cross-entropy loss function is mainly used for segmentation or classification problems. The loss is output by comparing the segmentation map and ground truth generated by the model proposed in this study. The cross-entropy loss function is expressed as

$$\text{Cross Entropy} = -\sum_k t_k \, \log_c (y_k) \tag{6}$$

In Equation (6), *k* is the *k*-th element of the training data, *t* is ground truth, and *y* is the output of the model.

## 4. Experiments and Results

### 4.1. Experiment Environment

To find out how well the proposed model works, we performed segmentation using wafer maps that include both single and mixed defects. We carried out all experiments on a GTX 1080Ti GPU with Intel Core i7-8700K CPU, 12 GB memory, and 16 GB RAM, and used a Keras open-source library based on Tensorflow. Table 1 summarizes the system specification.

**Table 1.** System specification.

| Hardware Environment | Software Environment |
| --- | --- |
| CPU: Intel Core i7-8700k, 3.7 Ghz, Six-core | Window |
| twelve threads 16 GB | Tensorflow 2.0 |
| GPU: Geforce GTX 1080Ti | Python 3.7 |

### 4.2. Dataset

#### 4.2.1. Single Defect

The dataset we used in this study, WM-811K, is a large, publicly available wafer map dataset, with 811,457 wafer maps collected from 46,393 lots [32]. The defect classes consist of Center, Donut, Edge-Loc, Edge-Ring, Loc, Scratch, Random, and Near-full. In this study, we used single defects as training data. To balance the training data, we randomly extracted 400 from each class. Afterwards, we increased the amount of data by data augmentation to improve the performance of the model. Figure 4 visually expresses the defects of WM-811K.

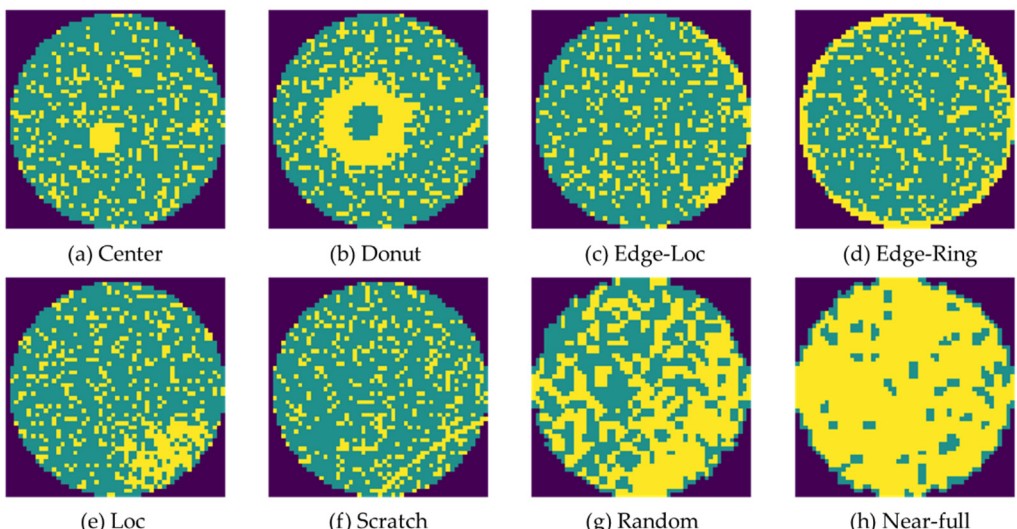

**Figure 4.** Single wafer map defect: (**a**) Center (C); (**b**) Donut (D); (**c**) Edge-Loc (EL); (**d**) Edge-Ring (ER); (**e**) Loc (L); (**f**) Scratch (S); (**g**) Random; (**h**) Near-full.

#### 4.2.2. Mixed-Type Defects

We intended to identify not only single defects but also mixed-type defects. Therefore, to assess the performance of the model, we used Mixed-type Wafer Defect Datasets provided by the Institute of Intelligent Manufacturing and Donghua University as a part of the test data [12]. This dataset is a wafer map dataset collected from a wafer fabrication plant, obtained by testing the electrical performance of each die on the wafer using test probes. We used mixed-type defects as test data to find out how well the proposed model could detect mixed defects. We used the two-mixed type, in which two defects are combined, and the three-mixed type, in which three defects are combined. However, we excluded the combination of Center and Donut from the experiment, because it may have many locations or overlapping parts; we also excluded Edge-Ring and Edge-Loc for the same

reason. For mixed-type defects, we randomly extracted 2400 pieces of data and used 1200 of them; these included both two-type and three-type defects. Figure 5 shows the mixed-type defects visually.

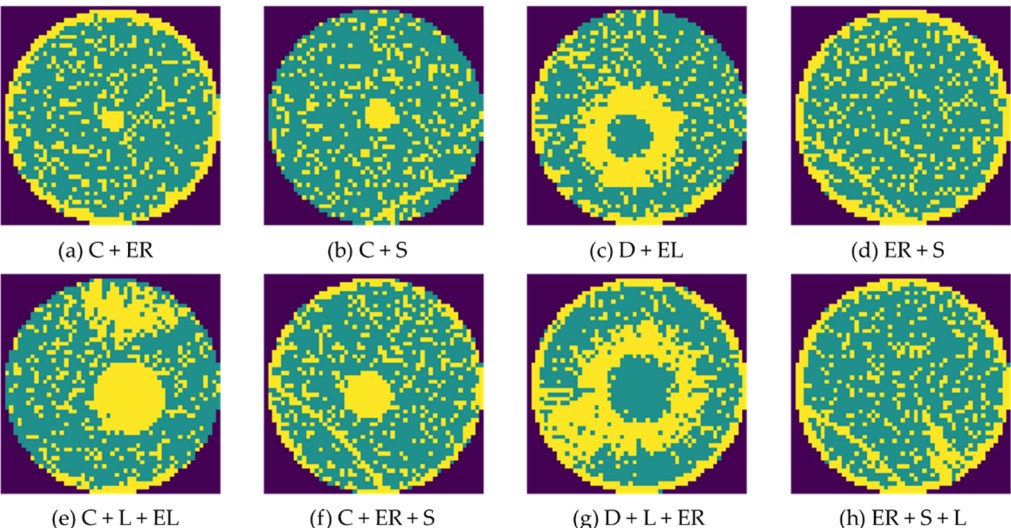

(a) C + ER  (b) C + S  (c) D + EL  (d) ER + S

(e) C + L + EL  (f) C + ER + S  (g) D + L + ER  (h) ER + S + L

**Figure 5.** Mixed-type wafer map defect: (**a**) Center + Edge-Ring; (**b**) Center + Scratch; (**c**) Donut + Edge-Loc; (**d**) Edge-Ring + Scratch; (**e**) Center + Loc + Edge-Loc; (**f**) Center + Edge-Ring + Scratch; (**g**) Donut + Loc + Edge-Ring; (**h**) Edge-Ring + Scratch + Loc.

### 4.3. Data Pre-Processing

#### 4.3.1. Defect Masking

Since the proposed model is based on U-Net, an image and a mask that is the ground truth of the image are required to train the model. In general, humans manually perform labeling using a program for image masking. These existing methods take a long time and require much manpower. To solve this problem, in this study, we used a method of automatically masking defects. Defects appearing on the wafer map are representations of a collection of defects in the die. Therefore, we performed masking in a way that distinguishes pixels that are connected to a certain number or more. For convenient masking, we converted the image to black and white and then masked it. Figures 6 and 7 show examples of black-and-white images and masking results for single and mixed defects.

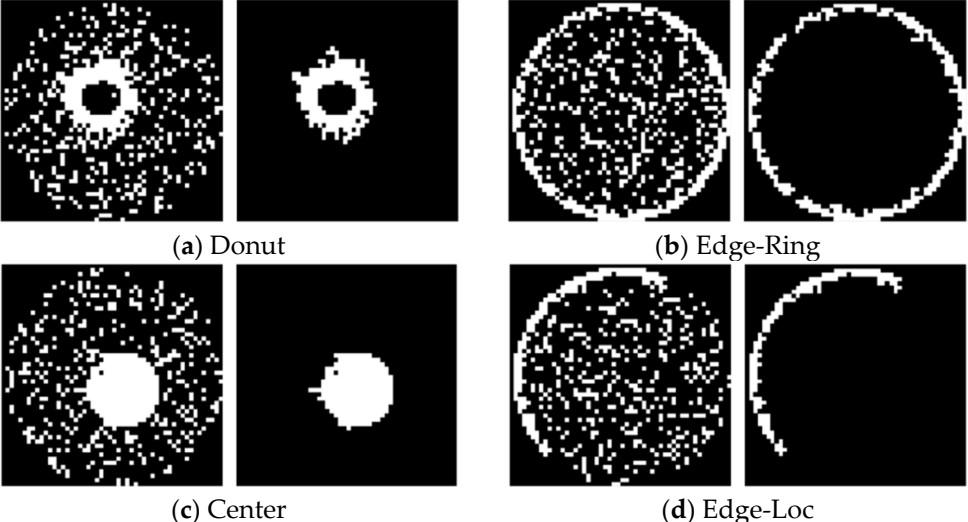

(**a**) Donut  (**b**) Edge-Ring

(**c**) Center  (**d**) Edge-Loc

**Figure 6.** Results of single defect masking: (**a**) Donut defect; (**b**) Edge-Ring defect; (**c**) Center defect; (**d**) Edge-Loc defect.

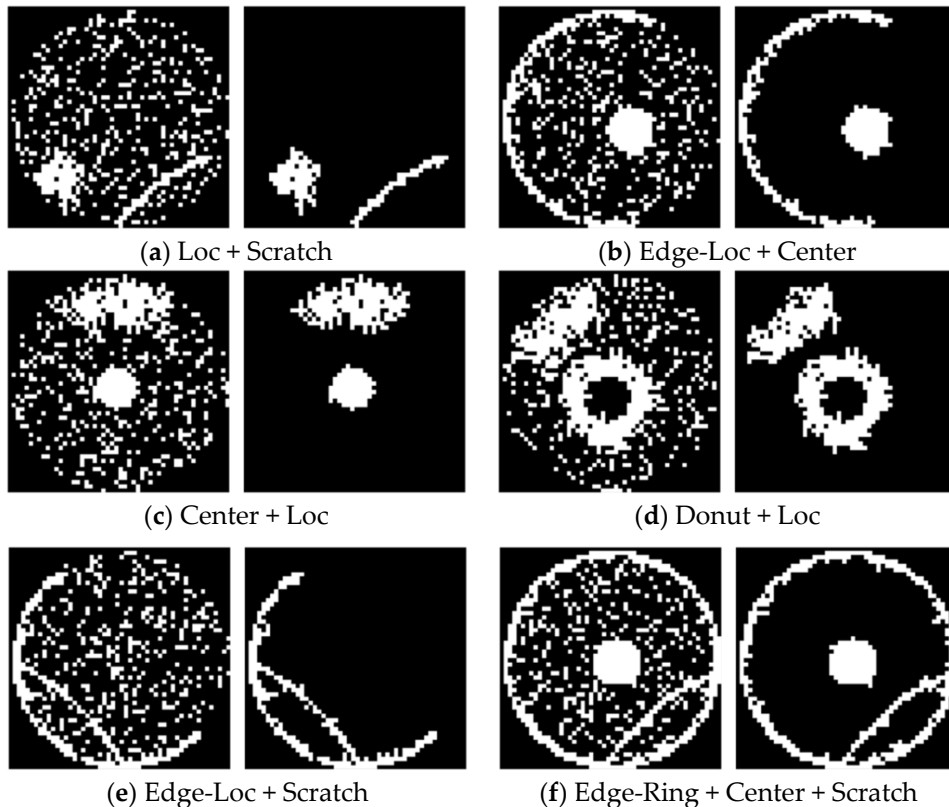

(**a**) Loc + Scratch  (**b**) Edge-Loc + Center

(**c**) Center + Loc  (**d**) Donut + Loc

(**e**) Edge-Loc + Scratch  (**f**) Edge-Ring + Center + Scratch

**Figure 7.** Results of mixed-type defect masking: (**a**) Loc + Scratch defect; (**b**) Edge-Loc + Center defect; (**c**) Center + Loc defect; (**d**) Donut + Loc defect; (**e**) Edge-Loc + Scratch defect; (**f**) Edge-Ring + Center + Scratch defect.

The image on the left is a black-and-white conversion of the original image, and the image on the right is the result of masking. We could accurately mask only the defect pattern without recognizing the surrounding defective die that was not a defect. For the Donut and Center, there was no big problem in recognizing the pattern of defects, although some parts protruding from the circle are also included. For the Edge-Ring and Edge-Loc, there were few defective dies on the edge, so breakage occurred.

For the mixed defects, the left image is a black-and-white conversion of the original image, and the right image is the result of masking. For the mixed defects, as shown in Figure 7, when the defects are separated, they are well distinguished. Both the two-mixed and three-mixed defects were mostly correctly distinguished. However, if the defects are closely connected or small, they can be recognized as a single defect. Therefore, for accurate verification, we excluded cases that could not be distinguished well from the test dataset.

### 4.3.2. Data Augmentation

We applied rotational data augmentation to improve the performance of the model. Data augmentation is a simple but easy way to improve the performance of deep neural networks. In this study, we adopted the image rotation method. First, we randomly extracted 100 single defects for each class, balanced the data, and then rotated them by 30 degrees. Figure 8 shows the results of rotation data augmentation, by which we were able to increase the size of the training dataset by 12 times.

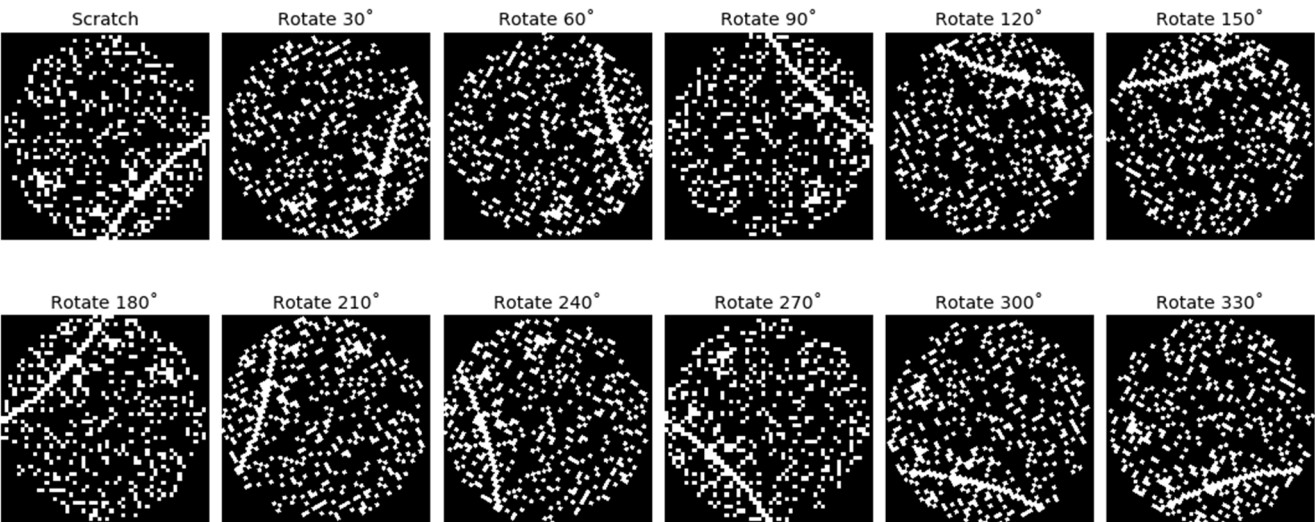

**Figure 8.** Result of rotation data augmentation.

*4.4. Evaluation Metrics*

To evaluate the model, we defined recall, precision, accuracy, and F1-score by using True Positive (TP), False Positive (FP), False Negative (FN), and True Negative (TN), which show the relationship between the answer presented by the model and the actual correct answer [33,34]. Precision is expressed as the ratio of the ones that are actually true to the ones that the model classifies as true.

$$\text{Precision} = \frac{TP}{TP + FP} \tag{7}$$

Recall is expressed as the ratio of those predicted by the model to be true to the total number of positive samples.

$$\text{Recall} = \frac{TP}{TP + FN} \tag{8}$$

Accuracy is the amount of correctly predicted data divided by the total amount of data. It is expressed as

$$\text{Accuracy} = \frac{|TP|}{|TP| + |FP| + |FN| + |TN|} \tag{9}$$

In multi-label classification, multiple classes are allowed differently from a single class, so the way to evaluate performance is used differently. In this study, we used multi-label accuracy, because mixed defects must be classified. Multi-label accuracy is the ratio of the total predicted data to the true predicted data. It is expressed as

$$\text{Multi} - \text{label Accuracy} = \frac{1}{W} \sum_{i=1}^{W} \frac{\hat{y}_i \cap y_i}{\hat{y}_i \cup y_i} \tag{10}$$

In Equation (11), $W$ is the total amount of wafer map data, $\hat{y}_i$ is the amount of data predicted by the model, and $y_i$ is the amount of data of actual defects.

F1-score is the harmonic average of precision and recall, and is expressed as

$$\text{F1} - \text{score} = \frac{\text{Precision} \times \text{Recall}}{\text{Precision} + \text{Recall}} \tag{11}$$

Intersection over Union (IoU) [35] is a metric frequently used for segmentation and object detection, and is expressed as

$$IoU = \frac{Target \cap Prediction}{Target \cup Prediction} \tag{12}$$

*4.5. Results*

4.5.1. Training Model

In this study, we used a U-Net model with residual attention blocks. After we performed defect masking to train the model, we obtained enough training data using rotation data augmentation. In addition, we could solve the data imbalance problem that resulted from the large differences in the ratio for each defect class. We used 9600 single-defect images as training data and added mixed defects to the test data to verify the model. However, Random and Near-full defects can interfere with detecting mixed defects, because the shape of the pattern is not clear, and most of them are composed of bad dies. Therefore, in this study, we excluded mixed defects containing both classes.

Figure 9 graphs the training loss of the model. The validation loss increases after 12 epochs, which means overfitting, so we adopted the model just before overfitting as the best model.

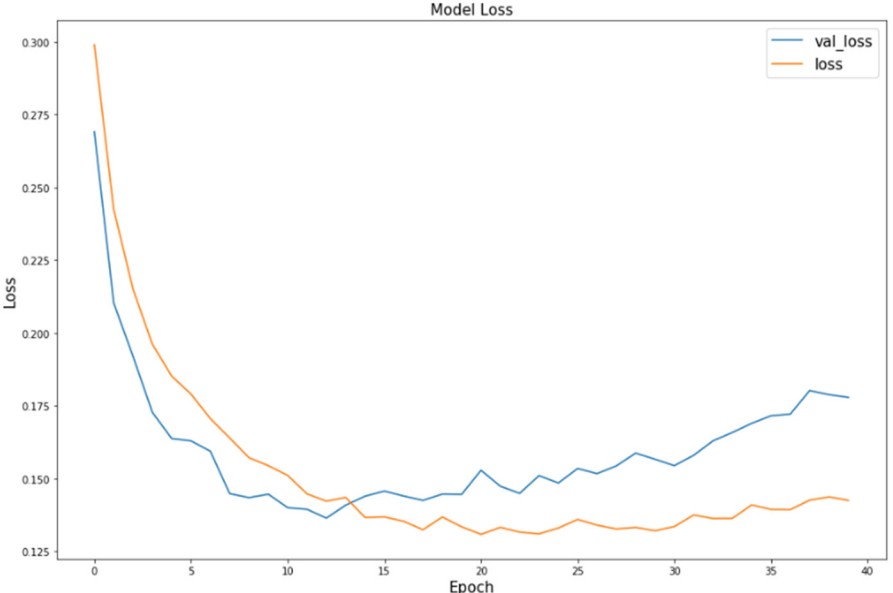

**Figure 9.** Training loss graph. The loss of the model according to the epoch is shown.

4.5.2. Single-Defect Result

Figure 10 shows the accuracy graph and confusion matrix for single defects. The test accuracy was more than 0.99 starting from epoch 5, after which it was possible to obtain results that maintained accuracy above a certain level.

Table 2 details the single-detection performance of the model. Accuracy for single defects reached 0.997. Because Center and Local defects have similar shapes, they may be misclassified depending on the defect location. A Scratch defect was sometimes predicted as an Edge-Loc defect. Nevertheless, since all single defects show an accuracy close to 1, it can be evaluated that our model accurately detects them.

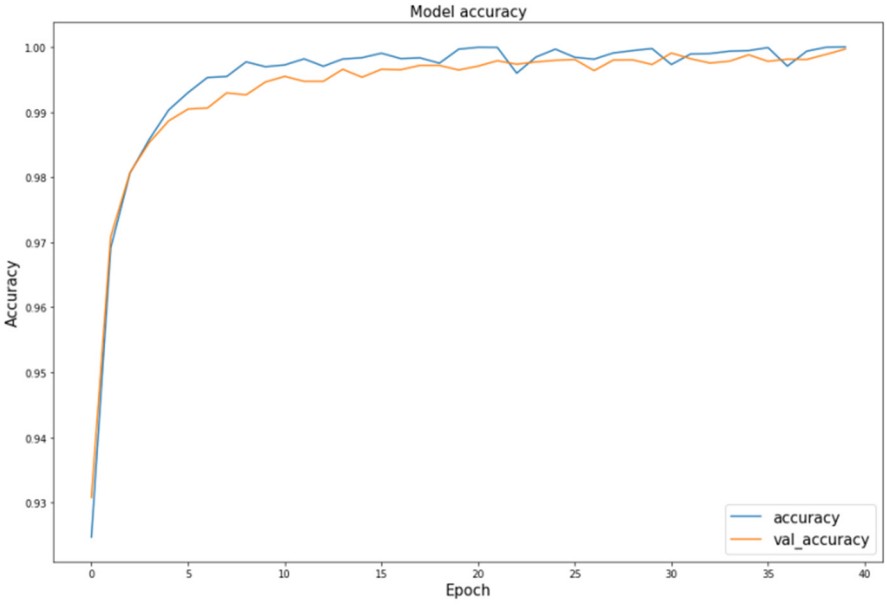

(**a**) Testing Accuracy Graph

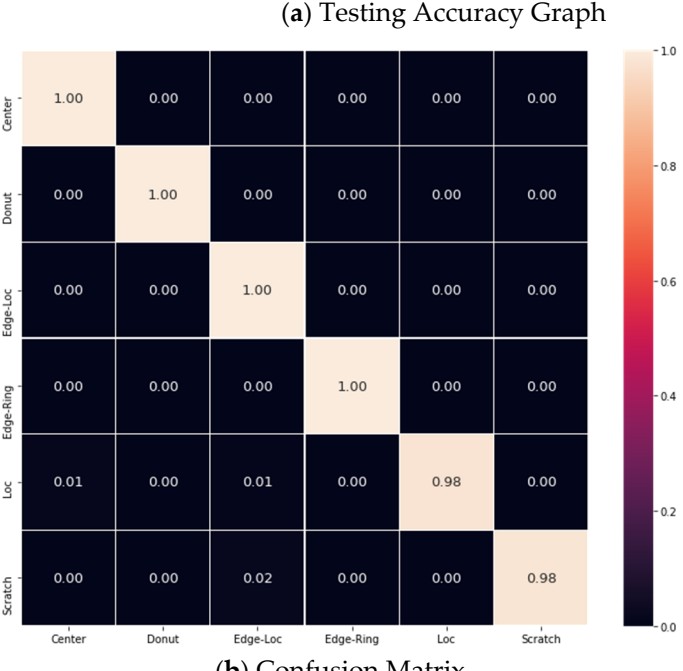

(**b**) Confusion Matrix

**Figure 10.** Testing results of single defects: (**a**) testing accuracy graph. The accuracy of the model according to the epoch is shown; (**b**) normalized confusion matrix (Center, Donut, Edge-Loc, Edge-Ring, Loc, Scratch).

**Table 2.** Single testing result.

| Defect Type | Accuracy | F1-Score | IoU |
|---|---|---|---|
| Center | 1.000 | 0.987 | 0.742 |
| Donut | 1.000 | 1.000 | 0.721 |
| Edge-Loc | 1.000 | 0.974 | 0.650 |
| Edge-Ring | 1.000 | 1.000 | 0.686 |
| Loc | 0.995 | 0.976 | 0.712 |
| Scratch | 0.987 | 0.982 | 0.720 |

Figure 11 shows some examples of segmentation results of single defects. It consists of the original image, ground truth, and result of segmentation for each defect. You can see that it accurately predicts a single defect when compared to the ground truth mask. Because ground truth acts as a label, the segmentation result is output as being as similar to ground truth as possible. Because the masking operation was performed automatically, there were some parts that did not match the defect, but there was no problem in detecting the defect.

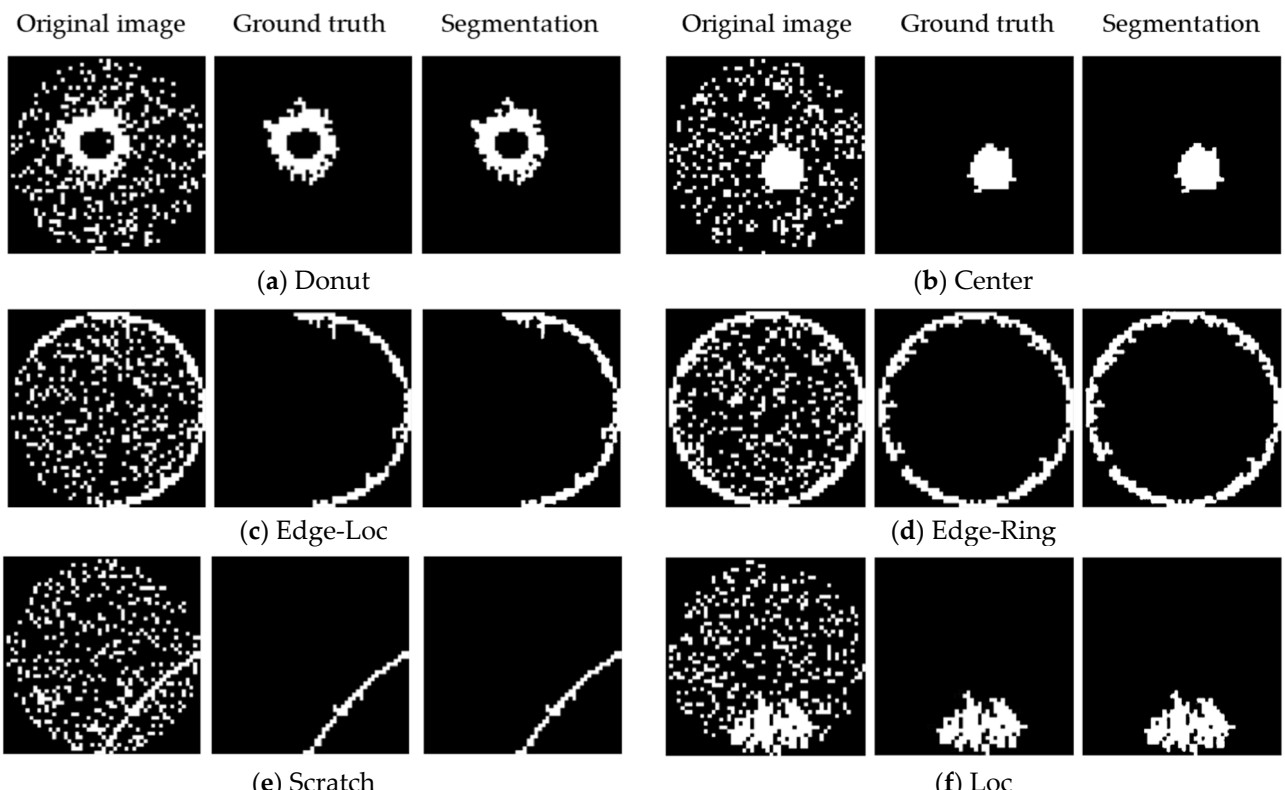

**Figure 11.** Segmentation Results in Single Defects: (**a**) Donut defect; (**b**) Center defect; (**c**) Edge-Loc defect; (**d**) Edge-Ring defect; (**e**) Scratch defect; (**f**) Loc defect.

### 4.5.3. Mixed-Type Defect Result

Figure 12 shows the accuracy graph and confusion matrix for mixed-type defects. The test accuracy was more than 0.97 by epoch 15, and we could obtain results that maintained high accuracy without reduction.

Table 3 details the detection performance for mixed-type faults. The multi-label accuracy for mixed-type defects reached 0.979. Sometimes, a mixed-type defect was judged to be a single defect when it had a combination of defects with similar morphologies or a combination of defects that mostly overlapped in positions. For this reason, the performance of the model was lower than for a single defect.

**Table 3.** Mixed-type testing result.

| Defect Type | Accuracy | F1-Score | IoU |
|---|---|---|---|
| Two-types Mixed | 0.979 | 0.982 | 0.645 |
| Three-types Mixed | 0.962 | 0.953 | 0.582 |

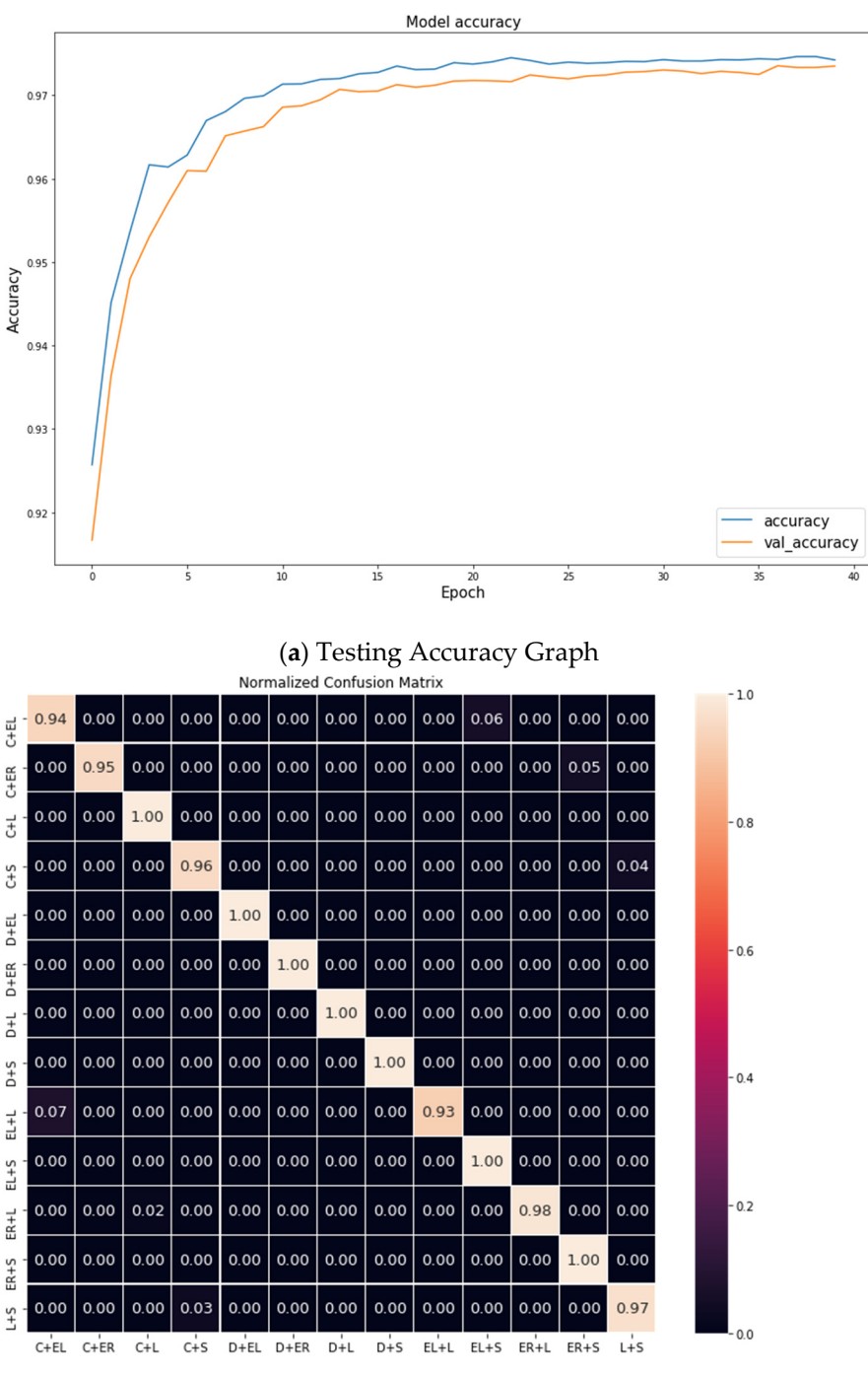

(**a**) Testing Accuracy Graph

(**b**) Confusion Matrix

**Figure 12.** Testing results of mixed-type defects: (**a**) testing accuracy graph. The accuracy of the model according to the epoch is shown; (**b**) normalized confusion matrix (C + EL, C + ER, C + L, C + S, D + EL, D + ER, D + L, D + S, EL + L, EL + S, EL + L, ER + S, L + S).

Figure 13 shows some examples of segmentation results of mixed defects. Mixed defects were also able to accurately predict most of the defects. However, it was difficult to detect all the defects when they overlapped by a certain ratio or more, for example, if Local and Edge-Loc defects overlap and are considered to be one Local defect, or if Scratches overlap with other defects and cannot be detected.

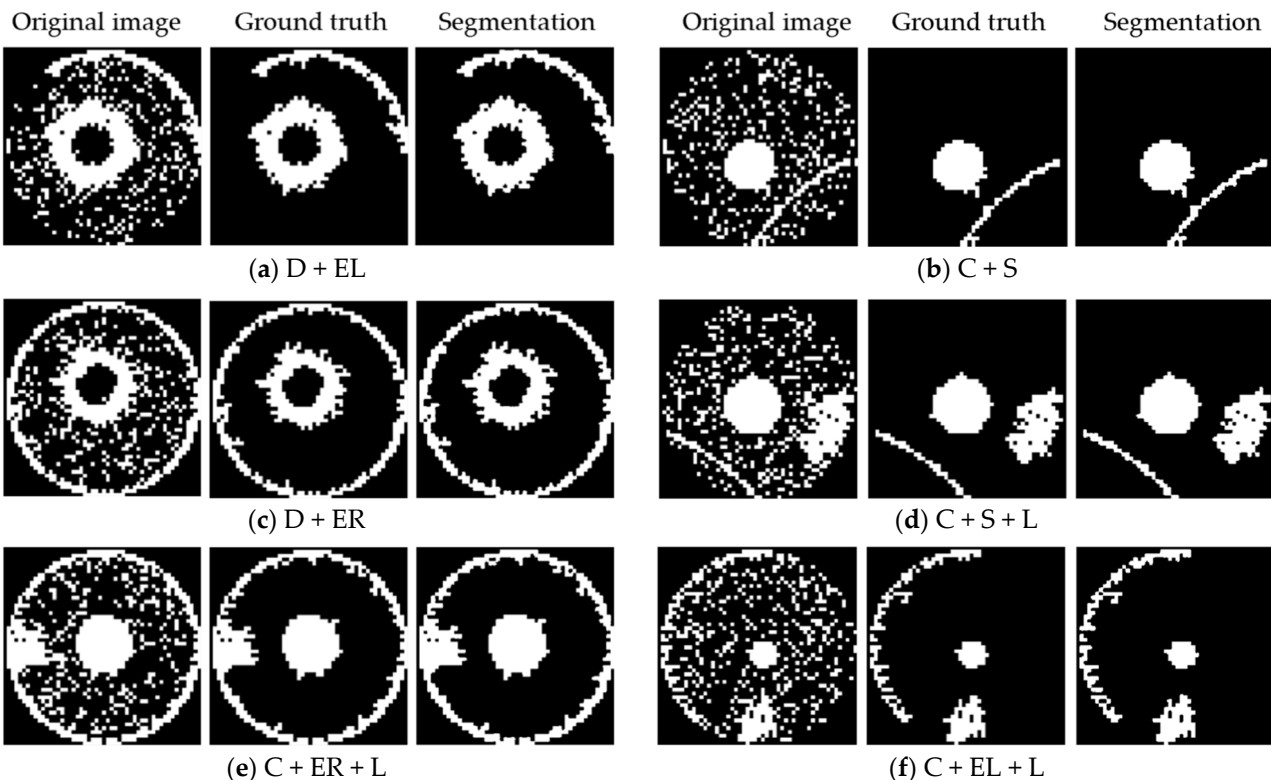

**Figure 13.** Segmentation results in mixed-type defect: (**a**) Donut + Edge-Loc defect; (**b**) Center + Scratch defect; (**c**) Donut + Edge-Ring defect; (**d**) Center + Loc + Scratch defect; (**e**) Center + Edge-Ring + Loc defect; (**f**) Center + Edge-Loc + Loc defect.

Finally, we compared the performance with that of existing studies in Table 4. Our model was superior to the basic models, U-Net and FCN, in all aspects. We obtained improved results because the residual attention block focused on the defect and considered the surrounding defective dies to be noise. The better performance can be confirmed even in comparison with previous studies. A few studies did not provide IoU, so we could not compare them, but for accuracy or F1-score, our model showed an improvement of 1% to 18%.

**Table 4.** Comparison with other models.

| Model | Accuracy | F1-Score | IoU |
|---|---|---|---|
| Our Model | 0.980 | 0.974 | 0.644 |
| U-Net | 0.958 | 0.868 | 0.482 |
| FCN | 0.942 | 0.841 | 0.464 |
| Wang et al. [12] | 0.826 | 0.824 | - |
| Kim et al. [13] | 0.962 | 0.962 | - |
| Ming-Chuan Chiu et al. [14] | 0.977 | 0.977 | 0.510 |

## 5. Conclusions

In this study, we propose an improved U-Net architecture using residual attention blocks, by means of which we were able to create an excellent feature map that focused on the defect information we wanted to find. Manual, subjective, and labor-intensive defect labeling is a very inefficient method. In this study, we solved the inefficiency problem by using automatic masking. By securing sufficient training data by using rotation data augmentation, we improved the performance of the model. This allowed us to detect not only single defects but also mixed defects. Our proposed method provided better performance than did the existing method. Accuracy was 0.980, F1-score 0.974, and IoU

0.644. These results will help engineers identify the cause of a problem by providing the exact location of the fault. Through this study, we were able to detect mixed defects with only a single defect, and the automatic defect masking technique was able to reduce unnecessary manpower and time. This can save the labor of existing workers and provide accurate defect detection performance.

We evaluated our model on two wafer map datasets, but further validation of the actual dataset can be considered in future studies. In addition, methods such as transfer learning can be used to improve the learning ability of the model. We plan to conduct future research focusing on reducing the weight of the model and improving its performance.

**Author Contributions:** Conceptualization, J.C. and J.J.; methodology, J.C.; software, J.C.; validation, J.C. and J.J.; formal analysis, J.C.; investigation, J.C.; resources, J.J.; data curation, J.C.; writing—original draft preparation, J.C.; writing—review and editing, J.J.; visualization, J.C.; supervision, J.J.; project administration, J.J.; funding acquisition, J.J. All authors have read and agreed to the published version of the manuscript.

**Funding:** This work was supported by the National Research Foundation of Korea (NRF) grant funded by the Korean government (MSIT) (No. 2021R1F1A1060054). And this work was supported by the National Research Foundation of Korea(NRF) grant funded by the Korea government(MSIT) (No. 2021R1F1A1060054) and the MSIT(Ministry of Science and ICT), Korea, under the ICT Creative Consilience Program(IITP-2021-2020-0-01821) supervised by the IITP(Institute for Information & communications Technology Planning & Evaluation). Corresponding author: Jongpil Jeong.

**Institutional Review Board Statement:** Not applicable.

**Informed Consent Statement:** Not applicable.

**Data Availability Statement:** Publicly available datasets were analyzed in this study. These can be found at (https://www.kaggle.com/qingyi/wm811k-wafer-map accessed on 26 February 2018) and (https://www.kaggle.com/co1d7era/mixedtype-wafer-defect-datasets accessed on 29 September 2020).

**Conflicts of Interest:** The authors declare no conflict of interest.

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
