# Peer review of "Improved U-Net with Residual Attention Block for Mixed-Defect Wafer Maps"

_applsci, doi:10.3390/app12042209_

Round 1

Reviewer 1 Report

Thanks for the work. In this work authors present an attention based U-Net for classification of wafer defects.

the authors claim three major contribution. 

1. Combining the residual attention block with the integrated residual block attention module with U-Net provides engineers with accurate segmentation results by focusing on the activation of the defect region.  

Unfortunately., this is a misleading contribution. There has been numerous papers on attention based U-net in literature that authors have not even referred to. In particular, three following papers are exactly the same topic and different implemenations of attention based U-Net. That begs the questions : a) why the authors didn’t refer to a single one of these papers? b) how they can justify the novelty of their work compared to such vast existing literature.  In my opinion, the better and more appropriate claim would have been: applying an attention guided u-net for classification of wafer defects.

https://link.springer.com/content/pdf/10.1007/s11042-020-10078-2.pdf https://arxiv.org/pdf/1804.03999.pdf

2. Through the rotation data augmentation technique and the defect masking technique, the performance of the model can be improved and human resources can  be reduced. 

this is not a contribution. Data augmentation has been used for many years, and routinely being used in literature. In fact with keras, there are existing routine to simply include augmentation, so not clear what is the contribution here!!

3. Evaluation of the single defect dataset and the mixed defect dataset was performed to show that the proposed model outperforms the basic model and the  existing studies

again this is already done in [14], so what do you mean by contribution? Simply testing the method and presenting the results is part of the paper writing process and not sure why it should be considered a major contribution.

there are several other issues with paper. Grammatical errors exist in the paper and paper need a major proofreading. There are places where authors yse conversational language instead of scientific writing!

the section explaining attention method is written very unclear and need significant improvement (specifically lines 125-142).

the captions in the figures has no information and not much details about them is provided in the text.

Many abbreviations (especially on the defect types) are used without saying what they are. 

Reviewer 2 Report

In this paper, the authors proposed application of the improved U-Net with residual attention block to evaluate mixed-type wafer maps. The contents seem interesting and useful, and there is no fatal error. So, I think that this paper can be published.

Reviewer 3 Report

Overall great work on coming up with an architecture that includes residual attention to improve classification performance. The following are comments/suggestions for the submitted manuscript.

  • Figures extracted from the python environment should be edited.
    • The font size of the x-axis and y-axis should be increased.
    • The legends need to include names instead of variable names from the python environment (this is preferred) other option is to include in the figure caption the meaning of each variable.
    • Confusion matrix variables also can be summarized in the caption or within a table in the manuscript.
  • I would add the significance of the work towards the main objective of that U-net in the conclusion. Currently its focused on performance of the U-net.

Round 2

Reviewer 1 Report

Thank for taking the time and editing the paper and responding to my comments.

While many of the concerns are addressed, I am still puzzled with some aspect of the work.

In particular, your Loss vs Epoch, as well as Accuracy vs Epoch curves are puzzling. 

1) Why the validation loss is lower than training loss at epoch 0 and overall the entire process before overfitting start?! Of course there are legitimate cases that this happens, but one of the reasons can be the sampling for validation may not be properly done. Authors should clarify this.

2) More importantly, the accuracy of the model at Epoch 0 on validation set (Figures 10a and 12a), is larger than the best accuracy you can achieve with U-Net and FCN in Table 4? Did you train U-Net and FCN? If you have trained them, how it is possible that a network with random initialization could produce higher accuracy than two state-of-the-art network that are trained on the data?? did you randomly initialized your network? Is Table 4 numbers are for the same test results?
